# Development and validation of new evaluation scale for measuring stroke patients' motivation for rehabilitation in rehabilitation wards

Taiki Yoshida[1,2,3], Yohei Otaka[1,4]*, Shin Kitamura[1,3], Kazuki Ushizawa[1,4], Masashi Kumagai[1], Yuto Kurihara[2], Jun Yaeda[5], Rieko Osu[6]

**1** Tokyo Bay Rehabilitation Hospital, Chiba, Japan, **2** Graduate School of Human Sciences, Waseda University, Saitama, Japan, **3** Faculty of Rehabilitation, School of Health Sciences, Fujita Health University, Aichi, Japan, **4** Department of Rehabilitation Medicine I, Fujita Health University School of Medicine, Aichi, Japan, **5** Graduate School of Comprehensive Human Science, University of Tsukuba, Tokyo, Japan, **6** Faculty of Human Sciences, Waseda University, Saitama, Japan

* otaka119@mac.com

**Data Availability Statement:** All relevant data are within the paper and its Supporting Information files.

## Abstract

### Objective

This study aimed to develop the Motivation in stroke patients for rehabilitation scale (MORE scale), following the Consensus-based standards for the selection of health measurement instruments (COSMIN).

### Method

Study participants included rehabilitation professionals working at the convalescent rehabilitation hospital and stroke patients admitted to the hospital. The original MORE scale was developed from an item pool, which was created through discussions of nine rehabilitation professionals. After the content validity of the scale was verified using the Delphi method with 61 rehabilitation professionals and 22 stroke patients, the scale's validity and reliability were examined for 201 stroke patients. The construct validity of the scale was investigated using exploratory factor analysis (EFA), confirmatory factor analysis (CFA), and item response theory analysis. Cronbach's alpha confirmed its internal consistency. Regarding convergent, discriminant, and criterion validity, Spearman's rho was calculated between the MORE scale and the Apathy Scale (AS), Self-rating Depression Scale (SDS), and Visual Analogue Scale (VAS), which rates the subjective feelings of motivation.

### Results

Using the Delphi method, 17 items were incorporated into the MORE scale. According to EFA and CFA, a one-factor model was suggested. All MORE scale items demonstrated satisfactory item response, with item slopes ranging from 0.811 to 2.142, and item difficulty parameters ranging from -3.203 to 0.522. Cronbach's alpha was 0.948. Regarding test-retest reliability, a moderate correlation was found between scores at the beginning and one month

**Funding:** T.Y.; JSPS KAKENHI Grant-in-Aid for Young Scientists (Start-up) (Grant no. 20K23271); https://kaken.nii.ac.jp/en/grant/KAKENHI-PROJECT-20K23271/ R.O.; Grantsin-Aid for Scientific Research (KAKENHI) on Innovative Areas (Hyper-Adaptability, 20H05482) fromthe Ministry of Education, Culture, Sports, Science and Technology (MEXT); https://kaken.nii.ac.jp/en/grant/KAKENHI-PUBLICLY-20H05482/ The funders had no role in study design, data collection and analysis, decision to publish, or preparation of the manuscript.

**Competing interests:** The authors have declared that no competing interests exist.

after hospitalization (rho = 0.612. p < 0.001). The MORE scale showed significant correlation with AS (rho = -0.536, p < 0.001), SDS (rho = -0.347, p < 0.001), and VAS (rho = 0.536, p < 0.001), confirming the convergent, discriminant, and criterion validity, respectively.

## Conclusions

The MORE scale was verified as a valid and reliable scale for evaluating stroke patients' motivation for rehabilitation.

## Introduction

Patients undergoing rehabilitation are required to be active participants in their exercise, and motivation is an important factor that influences their active participation. In rehabilitation, the motivation for rehabilitation can lead to an increase in patients' physical activity [1], improve their participation in their rehabilitation [2], and motivation could be associated with rehabilitation outcomes [3]. To accurately understand how patients' motivation for rehabilitation is related to rehabilitation outcomes, a scale for evaluating patients' motivation for rehabilitation is required.

Several scales for motivation for rehabilitation have been proposed for patients undergoing rehabilitation. However, each of these scales have some shortcomings. In self-determination theory [4, 5], widely known as the motivation theory, motivation is broadly classified into intrinsic motivation and extrinsic motivation. Extrinsic motivation involves performing a particular activity because it leads to a separable consequence; that is, the goal is separate from the activity itself. In contrast, intrinsic motivation involves performing a particular activity because it is interesting and enjoyable [4, 5]. According to this classification, rewards, including functional recovery and praise from medical staff and family members, can be categorized as extrinsic motivation, and patients' enjoyment of the rehabilitation itself can be classified as an intrinsic motivation. The motivation for traumatic brain injury rehabilitation questionnaire (MOT-Q) [6–9] consists of four extrinsic factors for patients' motivation. However, it has been reported that stroke patients' motivation for rehabilitation is influenced by broader factors other than those assessed using the MOT-Q [10]. Thus, MOT-Q may overlook some aspects of the factors that may affect motivation. The brain injury rehabilitation trust motivation questionnaire-self (BMQ-S) [8, 9, 11] is focused on only intrinsic motivation, which evaluates patients' motivation based on certain patients' personal traits. However, BMQ-S does not include intrinsic motivation-related items such as "I enjoy rehabilitation itself;" rather, it includes items about individual characteristics and traits. In short, it is unclear whether BMQ-S can specifically evaluate rehabilitation-related intrinsic motivation. Furthermore, the motivation for rehabilitation in patients undergoing rehabilitation in the hospital could be influenced by factors related to extrinsic motivation [10] thus, the evaluation scale consisting of intrinsic motivation cannot appropriately assess patients' motivation for rehabilitation.

The stroke rehabilitation motivation scale [12] has been adapted from the sports domain-related sports motivation scale [13] for use among stroke patients, and its items consist of three factors: amotivation, extrinsic motivation, and intrinsic motivation. However, it is unclear whether the motivation for sports on general person and the motivation for rehabilitation of patients with stroke are completely consistent. Therefore, the motivation scale in the sports domain could not correctly reflect the motivation for the rehabilitation of stroke patients.

The Pittsburg rehabilitation participation scale [14] evaluates patients' motivation based on their participation frequency and attitude for rehabilitation activities and has been developed for patients with various diseases. This scale assesses the patients' motivation based on the observations of medical staff. Regarding motivation and behavioral changes among stroke patients undergoing rehabilitation in hospitals, it has been pointed out that some patients have low levels of daily activity despite high motivation [10]. Such patients could be mislabeled as "low-motivated subjects" during observational evaluation by medical staff [15]. Namely, among some patients, the motivation label assigned to them by medical staff may differ from their actual motivation for rehabilitation. Thus, the Pittsburg rehabilitation participation scale [14] which evaluates patients' behavior cannot accurately reflect patients' motivation. The development of motivation evaluation items should include not only the views of medical staff but also those of patients. Thus, in such cases, it is desirable to use patient-reported outcome measures (PROMs), which are items based on qualitative data such as stroke patients' narratives.

PROMs lead to better communication and decision making between clinician and patients [16]. Generally, PROMs can be verified by using the Consensus-based standards for the selection of health measurement instruments (COSMIN) [17, 18]. However, scales for evaluating patients' motivation whose quality was verified based on COSMIN are still lacking. To develop a reliable scale for evaluating stroke patients' motivation for rehabilitation, it is necessary to validate the characteristics of this scale according to COSMIN.

In this study, we developed the Motivation in stroke patients for Rehabilitation scale (MORE scale) by referring to two types of factors (personal and social-relationship factors). They influence the patients' motivation for rehabilitation and the content of motivated behavioral change, which were revealed in our previous study [10]. Furthermore, the scale characteristics were verified using COSMIN. Hence, the MORE scale was appropriate for evaluating stroke patients' motivation for rehabilitation.

## Methods

This study was conducted according to COSMIN [17, 18]. The study protocol was approved by the Ethics Committee of Tokyo Bay Rehabilitation Hospital (No. 144) and the Ethics Committee of Waseda University (No. 2019–059). All participants provided written informed consent before participating in this study. Statistical analyses were performed with IBM SPSS Statistics 27.0 (IBM Corp., Armonk, NY), R (version 3.6.1) package "ltm," and "lavaan."

### Study setting

This study was conducted at convalescent rehabilitation wards called Kaifukuki Rehabilitation Wards (KRWs). KRW is the system for subacute rehabilitation in Japan and is covered by government medical insurance. In KRWs, patients undergo one-on-one intensive rehabilitation with therapists for around 2–3 hours every day. A typical schedule was 1 h in the morning and 1 or 2 h in the afternoon. Patients engage in self-training outside of rehabilitation sessions if indicated. The content of training with therapists and self-training is decided through discussions between rehabilitation professionals and patients.

### Developing items for MORE scale

The qualitative study that we previously conducted [10] revealed that the motivation of stroke patients admitted to KRWs was influenced by two factors—personal and social-relationship factors. Four categories of personal factors (patients' goals, experiences of success and failure, physical condition and cognitive function, and resilience) and three categories

of social-relationship factors (influence of rehabilitation professionals, relationship between patients, and patients' supporters) were included. Furthermore, the motivational status of stroke patients was shown to influence their behaviors, such as frequency of self-training and attitude toward activities in daily life. After referring to these previous findings [10], and the findings in another study on patients' views regarding motivation for rehabilitation [19], four rehabilitation professionals and the authors of the study [three occupational therapists (TY, MK, and SK) and a medical doctor (YO)] discussed and created an item pool in Japanese for the MORE scale. Then, a medical doctor, a nurse, a physical therapist, an occupational therapist, and a speech therapist who were not involved in the previous step held discussions to confirm the item pool validity. Thereafter, a final decision was made regarding the items, and the authors [TY, MK, SK, and YO] created a fixed item pool. To verify the content validity of the item pool, a two-round Delphi method [20] was conducted. Twenty-two stroke patients and 61 medical staff (20 physical therapists, 20 occupational therapists, and 21 nurses) were enrolled as participants. The participating patients were recruited through convenience sampling from patients hospitalized with hemorrhagic or ischemic stroke at the Tokyo Bay Rehabilitation Hospital between February and September 2017. The inclusion criteria were as follows: (1) first-time stroke; and (2) no physical or cognitive problems that could hinder the interview. The participating medical staff had more than five years of clinical experience in this hospital.

## Investigating validity and reliability

The next procedure involved verifying the validity and reliability of the MORE scale: the structural validity, the item response theory, internal consistency, and convergent, discriminant, and criterion validity. The participants were recruited from among a consecutive series of 527 patients who had been diagnosed with hemorrhagic or ischemic stroke and had been admitted at the Tokyo Bay Rehabilitation Hospital between October 2017 and March 2019. The inclusion criteria were as follows: (1) first-time stroke; and (2) no physical or cognitive problems that could hinder them from responding to the MORE scale. The participants' clinical characteristics were assessed using the Stroke Impairment Assessment Sets (SIAS) for assessing patients' physical function [21], and Functional Independence Measure (FIM) for assessing patients' functional disability [22].

**Structural validity.** The factor structure of the MORE scale was examined by performing exploratory factor analysis (EFA) and confirmatory factor analysis (CFA). EFA was performed to identify the number of factors and assess item factor loadings of MORE scale. The maximum likelihood estimation and promax rotation was performed. Kaiser-Meyer-Olkin measure of sampling adequacy and Bartlett's test of sphericity were used for assessing the suitability of the data for factor analysis. Kaiser-Meyer-Olkin value above 0.6 and Bartlett's test of sphericity p-value below 0.05 were appropriate for conducting a factor analysis [23]. To determine the appropriate number of factors in MORE scale, The Kaiser-Guttman rule was used [24, 25]. CFA was implemented to assess the fitness of the data to the factor structure extracted from the EFA. CFA was performed using maximum likelihood estimation. To investigate the models' goodness of fit, a number of statistics were used: chi-square, goodness of fit index [26], adjusted goodness of fit index [26], root mean square error of approximation [26], comparative fit index [26], Tucker-Lewis index [26], and standardized root mean square residual [26]. A good fit is defined as goodness of fit index greater than 0.95, adjusted goodness of fit index greater than 0.95, root mean square error of approximation less than 0.08, comparative fit index greater than 0.95, Tucker-Lewis index greater than 0.95, and standardized root mean square residual less than 0.08 [26]. We hypothesized that the MORE scale would have a three-

factor structure consisting of two motivation influencing factors (personal and social-relationship), and a behavioral change factor, similar to the results of our previous qualitative studies [10].

**Item response theory analysis.**   The item response theory (IRT) was used for investigating the properties of the items for MORE scale. We implemented the graded response model [27], which is appropriate for analyzing Likert-style item responses. IRT was used for estimating item slope parameters and item difficulty parameters in the MORE scale.

**Internal consistency.**   The Cronbach's alpha coefficient was evaluated for assessing internal consistency of the items in the MORE scale.

**Test-retest reliability.**   To evaluate the reliability of the MORE scale, the scale's results at the beginning of the hospitalization and its scores one month after the hospitalization were assessed, and the test-retest reliability was verified.

**Convergent, discriminant, and criterion validity.**   We examined the convergent validity based on the MORE scale's relationship with the Apathy Scale (AS) [28, 29], discriminant validity based on its relationship with the Self-rating Depression Scale (SDS) [30, 31], and criterion validity based on its relationship with the Visual Analogue Scale (VAS), which rates the subjective feelings of motivation. Depression and apathy are psychological problems associated with decreased motivation in rehabilitation practice. The symptoms of depression include symptoms such as lack of interest in events or activities that may be related to motivation; however, the main symptom is depressed mood [32–35]. Therefore, even if depression is correlated with motivation, which is the focus of this study, the correlation is expected to be weak. In contrast, since the symptoms of apathy include loss of motivation [32–35], the correlation with motivation is expected to be strong. For these reasons, we hypothesized that motivation would have a strong correlation with apathy and a weak correlation with depression. Thus, the AS was used to investigate the convergent validity, while the SDS was used to investigate the discriminant validity of the MORE scale. In addition, the VAS, which rates the subjective feelings of motivation, was used for evaluating the criterion validity since there lacked valid rating scales that could assess the motivation of stroke patients. Spearman's rho was evaluated to assess convergent and discriminative validity between the MORE scale and AS, SDS, and VAS. If the correlation between the MORE scale and AS is strong, it could be interpreted as evidence for convergent validity, and if the correlation between the MORE scale and SDS is weak, it could be interpreted as evidence for discriminant validity. Furthermore, if the correlation between the MORE scale and VAS is strong, it can be interpreted as evidence for criterion validity. Additionally, from the results of these psychological evaluations, we examined whether the MORE scale can specifically assess motivation toward rehabilitation.

AS assesses the apathy state, and consists of 14 items. Each item is scored on the following 4-point scale: 0, not at all; 1, slightly; 2, some; and 3, commonly. Total scores are from 0 to 42 points, with a higher score indicating more apathy state, and the cutoff value is 16 points [28]. The validity of the AS was established in stroke patients [29]. The SDS assesses depressive symptoms, and consists of 20 items. Each item is scored on the following 4-point scale: 1, rarely; 2, sometimes; 3, commonly; and 4, most of the time. Total scores range from 20 to 80 points, with a higher score indicating more depressive symptoms, and the cutoff value is 40 points [30]. The validity of the SDS was established in stroke patients [31]. In this study, the Japanese versions of the AS and SDS were used [36, 37]. The AS and SDS were adopted because these were self-rating scales similar to the MORE scale. Furthermore, these were easy and quick assessments in consideration of the participants' fatigue. VAS score of 100 implied a high motivation level, while a score of 0 indicated a low motivation level.

## Results

### Developing items for MORE scale

Nineteen items for evaluating patients' motivation were incorporated into an item pool. Eighty-three participants were included in the first survey round: 22 patients with stroke, 20 physical therapists, 20 occupational therapists, and 21 nurses. However, 80 (a physical therapist and two nurses dropped out) were included in the second survey round. The results of the two survey rounds showed that two of the 19 items did not obtain the 80% consensus among the participants (S1 Appendix). Table 1 shows the details of the MORE scale. The original Japanese version of the MORE scale is shown in S1 Table. The MORE scale contains 17 items, which were based on the following categories in our previous research [10]: four items (1,2,3,4) regarding the patients' goals, three items (11,12,13) regarding success and failure experiences, one item (14) regarding physical condition and cognitive function, two items (16,17) regarding resilience, four items (5,6,7,8) regarding the influence of rehabilitation professionals, one item (9) regarding relationships between patients, one item (10) regarding patients' supporters, and one item (15) regarding patients' behavior changes. In our previous study, patients' goals, success and failure experiences, physical condition and cognitive function, and resilience were based on the personal factors that influenced patients' motivation [10]. The influence of rehabilitation professionals, relationships between patients, and patients' supporters were based on the social relationship factors that influenced patients' motivation [10]. Each item of the MORE scale was evaluated using Likert scale. Considering the participants' fatigue, we selected the seven-point scale, which is known as the minimum optimal number on a Likert scale [38]. It was rated as follows: 1, Strongly disagree; 2, Disagree; 3, Somewhat disagree; 4, Neither agree nor disagree; 5, Somewhat agree; 6, Agree; and 7, Strongly agree.

**Table 1. Mean score of the MORE scale.**

| Item | | Mean (SD) n = 201 | | | Percentage of respondents "Strongly agree" |
|---|---|---|---|---|---|
| 1 | I want to participate in rehabilitation for my goals. | 6.3 | ( 1.0 ) | | 58.7 |
| 2 | I do not want to be discharged until I achieve my recovery objectives. | 5.8 | ( 1.3 ) | | 38.3 |
| 3 | I want to train in order to regain my role in my home and our society. | 5.9 | ( 1.3 ) | | 43.3 |
| 4 | I am able to make efforts to achieve my goal. | 6.0 | ( 1.1 ) | | 38.8 |
| 5 | I want to work hard to meet my therapists' expectations. | 6.2 | ( 1.0 ) | | 49.8 |
| 6 | I want to use the abilities I regained from the rehabilitation process in my daily life. | 6.2 | ( 0.9 ) | | 45.3 |
| 7 | I share my daily exercise target with my therapist on a daily basis. | 6.0 | ( 1.0 ) | | 40.3 |
| 8 | Alternations of daily rehabilitation plans propel me to participate more. | 6.0 | ( 1.0 ) | | 36.8 |
| 9 | I was encouraged by other patients' efforts. | 5.9 | ( 1.2 ) | | 39.3 |
| 10 | I want to participate in rehabilitation for the sake of my friends and family. | 6.2 | ( 1.0 ) | | 50.7 |
| 11 | I feel my body functions (such as body movement) improve on a daily basis. | 5.9 | ( 1.1 ) | | 34.3 |
| 12 | I would like to keep practicing so that I can regain my ability to perform lost/unexecuted daily activities. | 6.1 | ( 1.0 ) | | 42.3 |
| 13 | I want to try several different exercises/practices. | 5.9 | ( 1.1 ) | | 37.8 |
| 14 | I want to undergo the rehabilitation, even if I feel some pain and/or numbness. | 5.9 | ( 1.1 ) | | 38.8 |
| 15 | I want to train by myself in addition to usual supervised training. | 5.9 | ( 1.1 ) | | 32.8 |
| 16 | I think I must actively participate in rehabilitation. | 6.1 | ( 1.0 ) | | 45.3 |
| 17 | I think rehabilitation is essential for recovering from diseases and disabilities. | 6.4 | ( 0.9 ) | | 61.2 |
| Total score | | 102.8 | ( 13.5 ) | | |

## Investigating validity and reliability

**Participants' characteristics.** Participants of this study were selected from a consecutive series of 527 patients. According to inclusion criteria, 201 patients were included in this study. Table 2 shows participants' characteristics.

**Structural validity.** Table 1 shows the mean score and standard deviation for each item of the MORE scale. For all items, the percentage of respondents who answered "Strongly agree" was more than 15% of the total responses in all items; this suggested a ceiling effect for all the items [39, 40]. Floor effects were not observed in any of these items. Kaiser-Meyer-Olkin measure of sampling adequacy was 0.935, and Bartlett's test of sphericity gave a p-value <0.001, indicating a reasonable value for the factor analysis. The Kaiser-Guttman rule suggested that the MORE scale has a single factor structure (Eigenvalue = 9.11, variance extracted = 53.6%). Table 3 shows the factor loadings of each item. Factor loadings of the items ranged from 0.597 to 0.865. Given that the EFA could suggest a one-factor model, the specified model had just one latent factor (motivation for rehabilitation). As the results of CFA assume a one-factor model, the root mean square error of approximation as a measure of model fit was above 0.8, which was not a good model fit. In addition, the goodness of fit index, adjusted goodness of fit index, comparative fit index, and Tucker-Lewis index did not meet the criteria of a good model fit (chi-square = 426.6, df = 119.0 p<0.001; goodness of fit index = 0.803; adjusted goodness of fit index = 0.746; root mean square error of approximation = 0.114; comparative fit index = 0.871; Tucker-Lewis index = 0.852; root mean square error of approximation = 0.053). The results of the exploratory factor analysis indicated a one-factor structure, and the factor loadings for each item were at least 0.4 [41]. Therefore, factor analysis assuming other model structures was not conducted in the confirmatory factor analysis. Furthermore, the MORE scale was constructed without deleting any of the items because referring to the scores of each item makes it possible to examine an intervention method to motivate the participants.

**Item response theory analysis.** For items 4, 5, 6, 7, 8, 10, 11, 12, 13, 14, 15, 16, and 17, no participant responded with: 1, Strongly disagree; 2, Disagree; and 3, Somewhat disagree. Therefore, to conduct the analysis, we integrated points 1, 2, and 3 into one rating. Table 4 shows the results of the item slope parameters (alpha) and the item difficulty parameters

**Table 2. Participants' characteristics.**

| Characteristics | n = 201 | | | |
|---|---|---|---|---|
| Age, y. mean (SD) | 65.4 | ( | 13.6 | ) |
| Gender, female. n (%) | 81 | ( | 40.3 | ) |
| Days from stroke onset to admission. mean (SD) | 30.0 | ( | 13.9 | ) |
| Lesion side, right; left; both | 92; 105; 4 | | | |
| Type of stroke, hemorrhage; infarction | 127; 74 | | | |
| Days from admission to conduct the assessment. mean (SD) | 9.4 | ( | 4.5 | ) |
| Stroke Impairment Assessment Set motor function. median (IQR) | | | | |
| Knee-mouth test | 4 | ( | 3–5 | ) |
| Finger-function test | 4 | ( | 2–5 | ) |
| Hip-flexion test | 4 | ( | 4–5 | ) |
| Knee-extension test | 4 | ( | 4–5 | ) |
| Foot-pat test | 4 | ( | 3–5 | ) |
| Functional Independence Measure. median (IQR) | | | | |
| Total score | 88 | ( | 74–100 | ) |
| Motor score | 61 | ( | 48–70 | ) |
| Cognitive score | 28 | ( | 25–32 | ) |

**Table 3. Results of factor loadings with EFA.**

| Item | Factor 1 |
|------|----------|
| 1 | .730 |
| 2 | .727 |
| 3 | .668 |
| 4 | .624 |
| 5 | .709 |
| 6 | .828 |
| 7 | .713 |
| 8 | .734 |
| 9 | .620 |
| 10 | .805 |
| 11 | .597 |
| 12 | .865 |
| 13 | .818 |
| 14 | .649 |
| 15 | .734 |
| 16 | .824 |
| 17 | .731 |

(beta). Overall, 17 items demonstrated satisfactory item response, with item slopes ranging from 0.811 to 2.142. Item difficulty parameters ranged from -3.203 to 0.522.

**Internal consistency.** Cronbach's alpha, which evaluates internal consistency was 0.948. The items of the MORE scale showed excellent internal consistency.

**Table 4. Item slope parameters and difficulty parameters.**

| Item | α | β1 | β2 | β3 | β4 |
|------|------|--------|--------|--------|--------|
| 1 | 1.507 | -2.725 | -2.279 | -1.384 | -0.413 |
| 2 | 1.253 | -2.28 | -1.71 | -0.565 | 0.266 |
| 3 | 1.182 | -2.467 | -1.829 | -0.879 | 0.104 |
| 4 | 1.007 | -3.009 | -1.989 | -0.968 | 0.264 |
| 5 | 1.667 | -2.751 | -2.098 | -1.206 | -0.133 |
| 6 | 1.847 | -2.721 | -2.023 | -1.144 | 0.004 |
| 7 | 1.284 | -3.02 | -2.09 | -0.967 | 0.188 |
| 8 | 1.309 | -3.203 | -2.001 | -0.854 | 0.308 |
| 9 | 1.085 | -2.521 | -1.756 | -0.949 | 0.238 |
| 10 | 1.778 | -2.581 | -1.896 | -1.143 | -0.185 |
| 11 | 0.811 | -3.034 | -2.286 | -0.784 | 0.522 |
| 12 | 2.142 | -2.378 | -1.805 | -1.1 | 0.106 |
| 13 | 1.679 | -2.381 | -1.668 | -0.779 | 0.251 |
| 14 | 1.152 | -2.678 | -1.956 | -0.884 | 0.246 |
| 15 | 1.239 | -2.483 | -1.792 | -0.789 | 0.489 |
| 16 | 1.904 | -2.399 | -2.066 | -1.031 | 0.021 |
| 17 | 1.528 | -3.037 | -2.268 | -1.488 | -0.488 |
| **Mean** | 1.434 | -2.686 | -1.971 | -0.995 | 0.105 |
| **SD** | 0.361 | 0.285 | 0.196 | 0.232 | 0.280 |

A: Item slope parameter β: Item difficulty parameter

**Test-retest reliability.** One hundred and eight stroke patients who could be assessed for motivation by the MORE scale on admission and after one month of hospitalization were included to investigate the test-retest validity of the MORE scale. The normality of the MORE scale scores was analyzed using the Shapiro-Wilk test, which showed p < 0.001, indicating no normality. Then, the test-retest reliability was analyzed using Spearman's rank correlation coefficient, the result of which was ρ = 0.612, p<0.001. The time interval may have caused the values to be lower than those that would have been obtained by retesting immediately afterward; however, moderate reliability was confirmed.

**Convergent, discriminant, and criterion validity.** The convergent and discriminant was confirmed using the correlations between the MORE scale and the AS, SDS, and VAS. Seventeen of the 201 participants did not complete the AS and SDS. Therefore, correlations were analyzed using the data of the remaining 184 participants. The average score of the AS was 11.8 (SD±7.4), SDS was 39.8 (SD±8.9), VAS was 84.0 (SD±16.5), and MORE scale was 102.5 (SD±13.8). The number of participants in an apathetic state was 55 (29.9%), in a depressive state was 94 (51.6%), and in both an apathetic and depressive state was 42 (22.8%). The MORE scale showed a negative correlation with AS (rho = -0.567, p<0.001) and SDS (rho = -0.347, p<0.001). It showed a positive correlation with VAS (rho = 0.536, p<0.001) (Figs 1–3).

Furthermore, we validated the properties of the MORE scale using the cutoff value of the AS and SDS, and the quartile of the MORE scale. Participants with MORE scale scores below the first quartile were considered to have relatively low motivation. Table 5 shows the results of the total score of the AS, SDS, and MORE scale. Among the participants, 11.9% of those who were evaluated as having a score more than the cutoff point for both AS and SDS (both apathy and depression positive) obtained scores that were more than the third quartile for the MORE scale. However, 6.5% of participants who were evaluated as having a score below the cutoff point for both AS and SDS (both apathy and depression negative) were evaluated as having scores below the first quartile for the MORE scale. Thus, the results show that there were a certain number of participants who scored below the cutoff point for both depression and apathy, but had low motivation for rehabilitation. Conversely, there were a certain number of participants who scored above the cutoff point for both depression and apathy, but had high motivation for rehabilitation.

## Discussion

In this study, a new evaluation scale was developed for measuring stroke patients' motivation for rehabilitation (MORE scale) by referring to the factors extracted from their own narratives that influenced their motivation. The properties of the MORE scale were evaluated according to COSMIN. The results showed that the MORE scale was an appropriate scale for evaluating stroke patients' motivation for rehabilitation, and could specifically assess the motivation rather than depression and apathy.

It has been reported that stroke patients' motivation for rehabilitation in convalescent hospitals can be influenced by personal and social relationship factors, and could affect their behavioral changes [10]. All MORE scale items were developed by referring to two types of factors (personal and social-relationship factors) that influence patients' motivation for rehabilitation and the content of motivated behavioral change [10]. 10 items related to personal factors (patients' goal factors, factors regarding success and failure experiences, factors regarding physical and cognitive conditions, and factors regarding resilience), six items related to social relationship factors (factors regarding rehabilitation professionals, factors regarding patient relationships, and factors regarding patients' supporters), and one item related to patients' behavioral changes. Therefore, we initially hypothesized that the 17 MORE scale items could

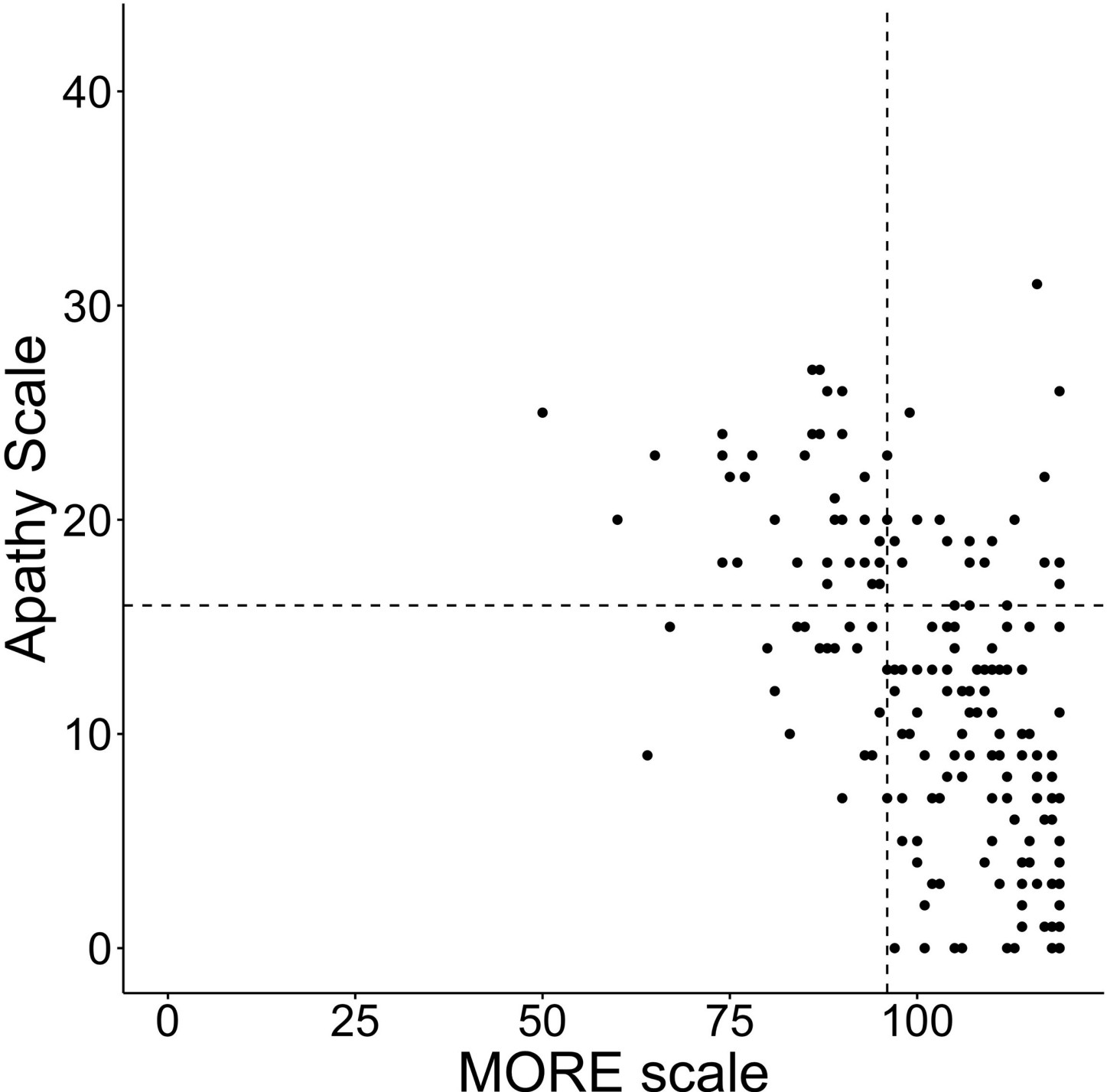

**Fig 1. Total scores of the MORE scale and the AS.** Each line on the X axis represents the first quartile of the MORE scale and Y axis represents the cutoff points of each scale.

be categorized into three factors corresponding to the eight categories of EFA. However, the EFA results indicated a one-factor structure, and the Cronbach's alpha coefficient value was very high. This could be caused by the ceiling effect in many items. The participants were

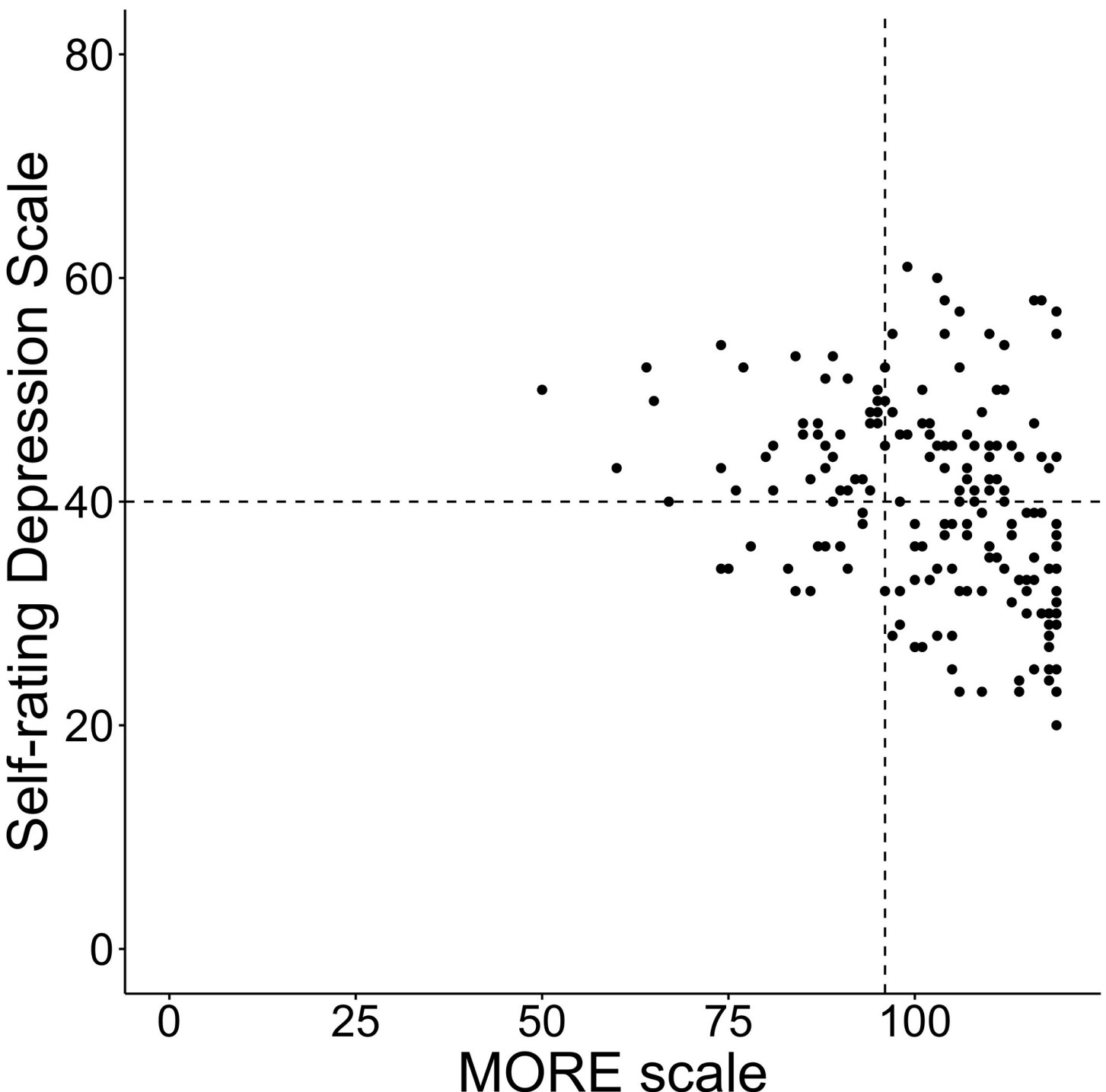

**Fig 2. Total scores of the MORE scale and the SDS.** Each line on the X axis represents the first quartile of the MORE scale and Y axis represents the cutoff points of each scale.

undergoing rehabilitation at a convalescent rehabilitation hospital, which provided intensive rehabilitation; furthermore, evaluation regarding psychological aspects was conducted early after the patients' hospitalization. Therefore, some participants may have had over-inflated

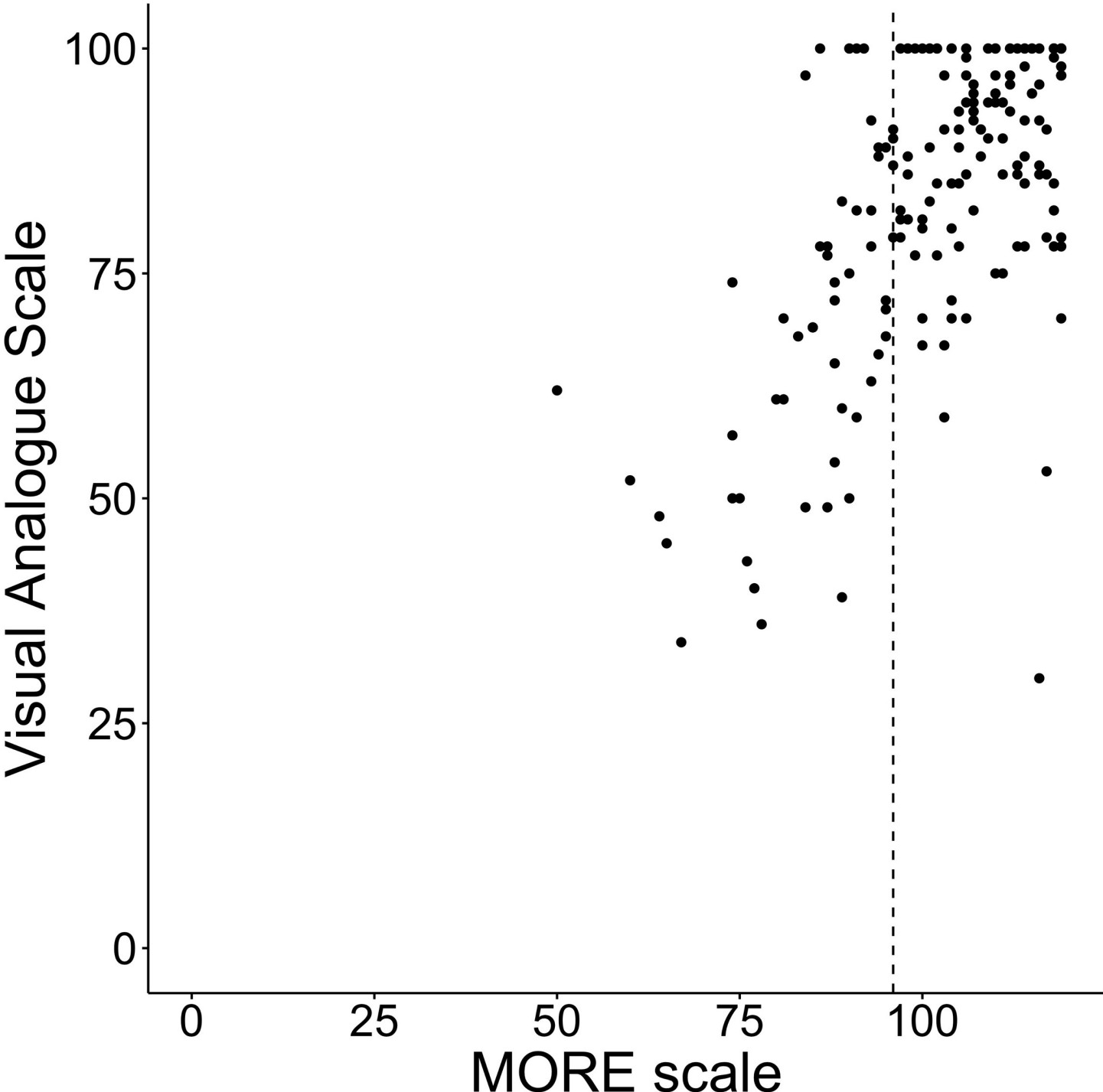

**Fig 3. Total scores of the MORE scale and the VAS.** Each line on the X axis represents the first quartile of the MORE scale and Y axis represents the cutoff points of each scale.

expectations regarding the rehabilitation. To clarify this point, further investigation is required to evaluate the time point when the patients themselves understand and agree to the realistic rehabilitation goals presented by the medical staff.

**Table 5. Classification of participation between MORE scale and status of apathy and depression.**

| | Apathy (-), Depression (-) | Apathy (+), Depression (+) |
|---|---|---|
| **MORE scale** | | |
| < 1st quartile | 5 (6.5%) | 20 (47.6%) |
| 1st– 3rd quartile | 41 (53.2%) | 17 (40.5%) |
| > 3rd quartile | 31 (40.3%) | 5 (11.9%) |
| **Total** | 77 (100.0%) | 42 (100.0%) |

Unlike in our hypothesis, this study's factor analysis, a one-factor structure was indicated. Thus, it is impossible to separate the motivation-related factors from the relevant item. However, this study found that there was a correlation between the VAS and the total score on the MORE scale. It was possible that the MORE scale measure motivation for rehabilitation in general. Furthermore, by referring to each item's scores, it became possible to understand, in detail, the categories that required motivational care. The category "goal setting," which the items 1,2,3, and 4 referred to, have been reported to be related to the improvement of daily living activities [42, 43]. "Pain" in Item 14, which is included in the category "physical condition and cognitive function", has been reported to have a negative effect on FIM improvement [44]. Similarly, the category "success experience", which are items 11, 12, and 13, and the category "resilience" (resilience against obstacles), which are items 16 and 17, were related to stroke patients' functional improvement [45, 46]. Furthermore, it has been reported that the category "influence from supporters," including professionals and family members, in items 5,6,7,8,9, and 10, can affect stroke patients' motivation for rehabilitation [15, 47–50]. Thus, referring to score composition according to each item could produce useful information for effectively planning rehabilitation. Furthermore, using the MORE scale to evaluate patient motivation could help to prevent any mislabeling of stroke patients' motivation in medical staff's observational evaluations. Aged stroke patients undergoing rehabilitation may have few emotional expressions (e.g., facial expressions) [10] and are often inactive; they are thus at risk of being mislabeled "unmotivated" by medical staff [15]. Using the MORE scale to evaluate motivations can help us reduce mislabeling and thus lead to more appropriate care practices.

The results based on IRT revealed strong support for a good item response on the MORE scale. The item difficulty parameter was relatively low. Thus, the MORE scale may have better discrimination ability among stroke patients reporting lower motivation for rehabilitation. The MORE scale showed moderate correlations with AS, SDS, and VAS. Loss of motivation has been reported as being closely related to and overlapping with apathy and depression [32–35]. Therefore, the observed correlations were considered reasonable. This study also examined whether the study participants were motivated with reference to the quartiles of the MORE scale. Consequently, we found a group of participants who had scores that were evaluated as having scores over the first quartile for the MORE scale and at the same time had scores that were above the cutoff point in AS or SDS (i.e., apathy or depression positive). Although the results of the motivation for rehabilitation evaluation among stroke patients were similar to those for apathy and depression in some aspects, apathy and depression did not necessarily correspond to motivation. The results showed that the MORE scale can evaluate patients' motivation for rehabilitation specifically, regardless of depression and apathy.

This study had several limitations. First, this study recruited participants from a single institution, and the participants had relatively high functional levels (as measured using FIM). In the future, it may be necessary to examine the MORE scale by conducting a large-sample survey, which should include participants with relatively low functional levels in daily life at other facilities. Second, this study's results were obtained only at one time-point during

hospitalization. However, the psychological status of patients is often expected to change during the hospitalization period. In the future, it will be possible to clarify the scale's characteristics and patients' motivations in more detail by confirming the evaluation results not only at one time-point after hospitalization but also at multiple time-points, including when the patients' goals are shared with medical staff and before their discharge. Third, in this study, we conducted validation using the Japanese version of the MORE scale among stroke patients admitted to a Japanese hospital. Language validation is necessary to examine whether the Japanese version of the MORE scale could be adapted for use in other countries as well.

This study's results showed that the MORE scale can be a valid scale for assessing motivation for rehabilitation in patients with stroke. Valid scales for assessing motivation for rehabilitation in patients with stroke have been scarce; medical staff may mislabel patients who should be designated as "highly motivated" as "low-motivated". It is expected that use of the MORE scale for evaluating motivation in rehabilitation practice will reduce instances of mislabeling by medical staff. Thus, rehabilitation professionals can utilize the MORE scale as a useful tool to understand stroke patients' motivation, increase their motivation, and develop optimal rehabilitation plans.

## Supporting information

**S1 Appendix. Percentage of item consensus by Delphi method.** Details of first and second round Delphi method results. Item 10 and 14 were excluded from MORE scale because less than 80% of the participants' consensus.
(DOCX)

**S1 Table. The original Japanese version of the MORE scale.** The items of original Japanese version of MORE scale.
(DOCX)

**S1 Data. Data of participant characteristics, clinical measures, MORE scale, apathy scale, self-rating depression scale, and MORE scale.** Data described are as follows: Patients' no; Gender; Age; Type of stroke; Paretic side; Days from stroke onset to admission; Total hospitalization days; Days from admission to psychological scales evaluation; FIM score; SIAS motor function score; MORE scale score; Self-rating depression scale score; and Apathy scale score.
(XLSX)

## Acknowledgments

We thank all the participants in this study. Many thanks to the staff at the Tokyo Bay Rehabilitation Hospital for finding participants.

## Author Contributions

**Conceptualization:** Taiki Yoshida, Yohei Otaka.

**Data curation:** Taiki Yoshida.

**Formal analysis:** Taiki Yoshida, Yohei Otaka, Rieko Osu.

**Funding acquisition:** Yohei Otaka, Rieko Osu.

**Investigation:** Taiki Yoshida, Shin Kitamura, Kazuki Ushizawa, Masashi Kumagai.

**Methodology:** Taiki Yoshida, Yohei Otaka, Rieko Osu.

**Project administration:** Taiki Yoshida.

**Resources:** Taiki Yoshida, Shin Kitamura, Kazuki Ushizawa, Masashi Kumagai.

**Software:** Taiki Yoshida, Yuto Kurihara.

**Supervision:** Yohei Otaka, Jun Yaeda, Rieko Osu.

**Visualization:** Taiki Yoshida, Yohei Otaka, Rieko Osu.

**Writing – original draft:** Taiki Yoshida, Yohei Otaka, Rieko Osu.

**Writing – review & editing:** Taiki Yoshida, Yohei Otaka, Shin Kitamura, Kazuki Ushizawa, Masashi Kumagai, Yuto Kurihara, Jun Yaeda, Rieko Osu.

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
