## [Decision Letter · Decision Letter 0]

5 Nov 2021

PONE-D-21-29804New evaluation scale for measuring patients’ motivation for rehabilitationPLOS ONE

Dear Dr. Otaka,

Thank you for submitting your manuscript to PLOS ONE. After careful consideration, we feel that it has merit but does not fully meet PLOS ONE’s publication criteria as it currently stands. Therefore, we invite you to submit a revised version of the manuscript that addresses the points raised during the review process. In addition to the clarifications / changes required by the reviewers, the title of the manuscript would be greatly improved if you include both the clinical subject (stroke) and the primary method. This will help readers to more quickly understand the relevance of the manuscript. Having looked at the included data file I consider you to have met the journal's requirements for data sharing. 

We look forward to receiving your revised manuscript.

Kind regards,

Jason Scott

Academic Editor

PLOS ONE

Journal Requirements:

Additional Editor Comments:

Please update the title to include the clinical setting and study method

Reviewers' comments:

Reviewer's Responses to Questions

**Comments to the Author**

1. Is the manuscript technically sound, and do the data support the conclusions?

Reviewer #1: Yes

Reviewer #2: No

2. Has the statistical analysis been performed appropriately and rigorously? 

Reviewer #1: Yes

Reviewer #2: No

3. Have the authors made all data underlying the findings in their manuscript fully available?

Reviewer #1: Yes

Reviewer #2: No

4. Is the manuscript presented in an intelligible fashion and written in standard English?

Reviewer #1: Yes

Reviewer #2: Yes

5. Review Comments to the Author

Reviewer #1: It is an interesting topic to discuss. The title and abstract are interesting and easy to read. The introduction has explained the background of the study and the gap in the knowledge related to patients' motivation for rehabilitation scale and the shortcoming of previous proposed scales. The methods section is both clear and detail enough that it explained several steps of the MORE scale development and the scale test for its validity, reliability. The results section was written clearly and informatively. Data were presented in clear tables followed by ample explanation. . The discussion section was adequate and the results were discuss from several perspectives without being too wordy or overinterpreted. Conclusion section had concluded the paper properly

Reviewer #2: Summary

This is a well-designed study which deals with motivation during rehabilitation. Here are my suggestions

Major comments

It needs more explanation on the MORE. Is it stroke-specific or generic?

MORE seems to represent rehabilitation motivation, however the developments was only based on the stroke patients, thus the naming is preferred to be changed and target population could be specified.

The language might affect the results. For example, the validity and reliability of Japanese version of SDS and AS needs to be written in this paragraphs with references. Especially, these questionnaires related to psychometric properties are easy to be affected by the language. For example SRMS and Korean version of SRMS is different because there are different consistency. (Park M, Lee JY, Ham Y, Oh SW, Shin JH. Korean Version of the Stroke Rehabilitation Motivation Scale: Reliability and Validity Evaluation. Ann Rehabil Med. 2020;44(1):11-19. doi:10.5535/arm.2020.44.1.11) These concerns about language needs to be listed on the limitation.

Similarly, the English version of the MORE might be different from original version of the MORE. Thus, both the English version of the MORE and Japanese original version of the MORE needs to be listed on this manuscript and those things are better to be listed in the limitation.

Under the subtitle of convergent, discriminant, and criterion validity, discriminant and criterion validity was not enough. Please put more things on that topic. Also, topics regarding reliability was missed on this article.

There is a logical gap between the result and the discussion (The results showed that the MORE scale was an appropriate scale for evaluating stroke patients' motivation for rehabilitation, and could specifically assess the motivation rather than depression and apathy.) Especially, it failed to demonstrate structural validity, discriminant, and criterion validity as well as reliability. The results describes single factor model, however discussion says three factors (personal, social, behavioral). These descriptions does not come from the result, thus please provide the logical support.

Minor

There is inequity on the introduction of the various outcomes. Thus, there are too much expression about MOT-Q.

L97: based on “various categories”. Please specify the various categories.

L98 from medical staff but also narratives from stroke patients themselves [14]. It is difficult to understand the contents of this sentences and the reference 14.

L123: “from among” is strange to be read. Please fix it

L128: “the item response theory” seems to be better to be revised into “item response theory analysis” and it applies on the remaining sentences on this manuscript.

The authors mentioned “Intensive rehabilitation”, however it is not defined in the manuscript, thus it needs operational definition

L158, 159: The authors describe association between motivation and apathy, and apathy and depression. However, the goal of the present study was to find association of motivation and apathy, and motivation and depression. Thus those part needs to be revised.

L161: needs grammatical correction.

L167: It might be better to describe the reason of using SDS other than other outcome measures on the depression. Moreover, the reference on the validation and reliability of SDS is needed.

L183-186: It needs clarification and specification of the process grouping the items, why the whole items were grouped into personal factor, social relationship factor.

L187: Why the Likert scale is made up with 7-point scale?. Please state the reason.

Table 2:

Paretic side: 0 could be misunderstood , thus lesion side seems to be better to transfer the glimpse of the participants.

Foot-tap test rather than foot-pat test

L 204: Please specify “it”

P15: structural validity says the MORE scale is not indicative of acceptable model fit. Then, I guess more revision is needed in the process, however there was no trial to improve the validity.

P20 L12-18: It is difficult to understand the authors’ purpose of the table 6 and the sentences. I hope the authors emphasize the meaning of the table.

Table 6 :abbreviation of IQR needs to be described with full terminology. The row name IQR needs to be revised to convey the meaning of 1st ~ 3rd IQR.

Table 5: It seems to be strange to correlate AS total score, SDS total score, and VAS with each item of MORE scale. Please specify the reason of this process.

Discussion

Unlike in our hypothesis, this study's factor analysis, a one-factor structure was indicated.; In the introduction section, hypothesis about the factor is not found. Please describe it.

“The category “goal setting,” which the items 1,2,3, and 4 referred to, have been reported to be related to the improvement of daily living activities [38,39]. "Pain" in Item 14, which is included in the category "physical condition", has been reported to have a negative effect on FIM improvement [40]. Similarly, the category "success experience", which are items 11, 12, and 13, and the category "resilience" (resilience against obstacles), which are items 16 and 17, were related to patients’ functional improvement [41,42]. Furthermore, it has been reported that the category "influence from supporters," including professionals and family members, in items 5,6,7,8,9, and 10, can affect patients' motivation for rehabilitation [15,43-46]. “ These paragraphs seems to be strange, because the authors grouped each items and named it, without support by results. Lack of scientific evidence does not support the meaning of this article

“This study also investigated the cutoff point of the MORE scale”. There seems no description about the cutoff point of the MORE scale in the result section. Please amend it.

“The results showed that the MORE scale can evaluate patients’ motivation for rehabilitation specifically, regardless of depression and apathy.” needs evidence from results. If there is relevant result from this study, please describe it and explain with explanation.

6. PLOS authors have the option to publish the peer review history of their article (what does this mean?). If published, this will include your full peer review and any attached files.

Reviewer #1: **Yes: **Ninuk Dian Kurniawati

Reviewer #2: No

---

## [Author Response · Author response to Decision Letter 0]

4 Jan 2022

Reviewer Comments:

Comments from Reviewer #1:

It is an interesting topic to discuss. The title and abstract are interesting and easy to read. The introduction has explained the background of the study and the gap in the knowledge related to patients' motivation for rehabilitation scale and the shortcoming of previous proposed scales. The methods section is both clear and detail enough that it explained several steps of the MORE scale development and the scale test for its validity, reliability. The results section was written clearly and informatively. Data were presented in clear tables followed by ample explanation. The discussion section was adequate and the results were discuss from several perspectives without being too wordy or overinterpreted. Conclusion section had concluded the paper properly

Authors’ comments:

Thank you for your positive comments.

 

Comments from Reviewer #2:

Major comment 1

It needs more explanation on the MORE. Is it stroke-specific or generic? MORE seems to represent rehabilitation motivation, however the developments was only based on the stroke patients, thus the naming is preferred to be changed and target population could be specified.

Authors’ comments:

Thank you for your comment. Our previous studies on the MORE scale, which we referred to when creating the items, were conducted with stroke patients, and the current validation of the scale's characteristics was also conducted only with stroke patients. Therefore, to avoid any misunderstanding, the term “stroke patients" rather than "patients" was used in the revised manuscript. Furthermore, the name of the MORE scale has been revised as “Motivation in stroke patients for Rehabilitation scale." Furthermore, we updated the title to include the clinical setting and study method.

Major comments 2

The language might affect the results. For example, the validity and reliability of Japanese version of SDS and AS needs to be written in these paragraphs with references. Especially, these questionnaires related to psychometric properties are easy to be affected by the language. For example, SRMS and Korean version of SRMS is different because there are different consistency. (Park M, Lee JY, Ham Y, Oh SW, Shin JH. Korean Version of the Stroke Rehabilitation Motivation Scale: Reliability and Validity Evaluation. Ann Rehabil Med. 2020;44(1):11-19. doi:10.5535/arm.2020.44.1.11) These concerns about language needs to be listed on the limitation. Similarly, the English version of the MORE might be different from original version of the MORE. Thus, both the English version of the MORE and Japanese original version of the MORE needs to be listed on this manuscript and those things are better to be listed in the limitation.

Authors’ comments:

Thank you for your comment. We have cited as references the studies in which the validity and reliability of the Japanese versions of the SDS and AS have been verified. As for the language validity of the MORE scale, we have stated the necessity of language validation in the future as a limitation of the study. In addition, we have provided the original Japanese version of MORE scale in Supplementary Table 1.

P12, L243

In this study, the Japanese versions of the AS and SDS were used [36, 37].

36. Okada K, Kobayashi S, Aoki K, Suyama N, Yamaguchi S. Assessment of motivational loss in poststroke patients using the Japanese version of Starkstein’s apathy scale. Jps J Stroke. 1998;20(3):318-323. Doi: 10.3995/jstroke.20.318 (in Japanese)

37. Fukuda K, Kobayashi S. A study of a self-rating depression scale. Seishin Shinkeigaku Zasshi. 1973;10:673-639 (In Japanese).

P12, L254-

Table 1 shows the details of the MORE scale. The original Japanese version of the MORE scale is shown in Supplementary Table 1.

P26, L417–

Third, in this study, we conducted validation using the Japanese version of the MORE scale among stroke patients admitted to a Japanese hospital. Language validation is necessary to examine whether the Japanese version of the MORE scale could be adapted for use in other countries as well.

Major comment 3

Under the subtitle of convergent, discriminant, and criterion validity, discriminant and criterion validity was not enough. Please put more things on that topic. Also, topics regarding reliability was missed on this article.

Authors’ comments:

Thank you for your comment. In this study, we examined the validity of the MORE scale based on its relationship with other scales: the convergent validity was established based on its relationship with the Apathy Scale (AS), discriminant validity based on its relationship with the Self-rating Depression Scale (SDS), and criterion validity based on its relationship with the Visual Analogue Scale (VAS), which rates the subjective feelings of motivation. Depression and apathy are psychological problems that are associated with low motivation in rehabilitation clinical practice. Previous studies have reported that the symptoms of depression include a lack of interest in events or activities that may be related to motivation; however, the main symptom is depressed mood. Therefore, even if there is a correlation of depression with motivation, which is the focus of this study, the correlation is expected to be weak. In contrast, since the symptoms of apathy include loss of motivation, the correlation with motivation is expected to be strong. Thus, we hypothesized that the scores of the MORE scale would have a strong correlation with the AS and VAS and a weak correlation with the SDS. 

In the results, AS was moderately correlated with the MORE scale under convergent validity, SDS was weakly correlated with the MORE scale under discriminant validity, and VAS was moderately correlated with the MORE scale under criterion validity. These results are reasonable considering our hypothesis and previous studies. These findings have been included in the manuscript.

Further, by examining the relationship between depression, apathy, and motivation for rehabilitation based on the results of scores on the AS, SDS, and MORE scale, we examined whether the MORE scale could specifically assess stroke patients’ motivation for rehabilitation, regardless of depression or apathy. This point has been corrected.

For reliability, test-retest reliability was examined using the results of 108 evaluations at the time of admission and approximately one month later.

P11, L212-

We examined the convergent validity based on the MORE scale’s relationship with the Apathy Scale (AS) [32,33], discriminant validity based on its relationship with the Self-rating Depression Scale (SDS) [34,35], and criterion validity based on its relationship with the Visual Analogue Scale (VAS), which rates the subjective feelings of motivation. Depression and apathy are psychological problems associated with decreased motivation in rehabilitation practice. The symptoms of depression include symptoms such as lack of interest in events or activities that may be related to motivation; however, the main symptom is depressed mood [29-31]. Therefore, even if depression is correlated with motivation, which is the focus of this study, the correlation is expected to be weak. In contrast, since the symptoms of apathy include loss of motivation [29-31], the correlation with motivation is expected to be strong. For these reasons, we hypothesized that motivation would have a strong correlation with apathy and a weak correlation with depression. Thus, the AS was used to investigate the convergent validity, while the SDS was used to investigate the discriminant validity of the MORE scale. In addition, the VAS, which rates the subjective feelings of motivation, was used for evaluating the criterion validity since there lacked valid rating scales that could assess the motivation of stroke patients. Spearman’s rho was evaluated to assess convergent and discriminative validity between the MORE scale and AS, SDS, and VAS. If the correlation between the MORE scale and AS is strong, it could be interpreted as evidence for convergent validity, and if the correlation between the MORE scale and SDS is weak, it could be interpreted as evidence for discriminant validity. Furthermore, if the correlation between the MORE scale and VAS is strong, it can be interpreted as evidence for criterion validity. Additionally, from the results of these psychological evaluations, we examined whether the MORE scale can specifically assess motivation toward rehabilitation. 

P10, L205-: Methods section

Test-retest reliability 

To evaluate the reliability of the MORE scale, the scale’s results at the beginning of the hospitalization and its scores one month after the hospitalization were assessed, and the test-retest reliability was verified.

P20, L318–: Results section

Test-retest reliability 

One hundred and eight stroke patients who could be assessed for motivation by the MORE scale on admission and after one month of hospitalization were included to investigate the test-retest validity of the MORE scale. The normality of the MORE scale scores was analyzed using the Shapiro-Wilk test, which showed p < 0.001, indicating no normality. Then, the test-retest reliability was analyzed using Spearman's rank correlation coefficient, the result of which was ρ = 0.612. The time interval may have caused the values to be lower than those that would have been obtained by retesting immediately afterward; however, moderate reliability was confirmed.

Major comment 4

There is a logical gap between the result and the discussion (The results showed that the MORE scale was an appropriate scale for evaluating stroke patients' motivation for rehabilitation, and could specifically assess the motivation rather than depression and apathy.) Especially, it failed to demonstrate structural validity, discriminant, and criterion validity as well as reliability.

Authors’ comments:

Thank you for your comments. We have revised the descriptions on discriminant and convergent validity and criterion-related validity, as explained in the answer to Major comment 3. As for reliability, the description about test-retest reliability has been newly added based on Major comment 3.

Regarding the structural validity, the results of the exploratory factor analysis suggested a one-factor structure model, and the factor loadings for each factor were at least 0.4. Therefore, we concluded that there was little meaning in further verification on a structure. Furthermore, by referring to the score of each item of the MORE scale, we can obtain information on what type of interventions could influence patients’ motivation. For this reason, we did not delete any of the items on the MORE scale despite the scale having a one-factor structure. We have revised the manuscript as follows.

P16, L296–

The results of the exploratory factor analysis indicated a one-factor structure, and the factor loadings for each item were at least 0.4 [41]. Therefore, factor analysis assuming other model structures was not conducted in the confirmatory factor analysis. Furthermore, the MORE scale was constructed without deleting any of the items because referring to the scores of each item makes it possible to examine an intervention method to motivate the participants.

41. Costello AB, Osborne J. Best practices in exploratory factor analysis: four recommendations for getting the most from your analysis. Pract Assess Res Eval. 2005;10:7. doi: https://doi.org/10.7275/jyj1-4868

Major comments 5

The results describe single factor model, however discussion says three factors (personal, social, behavioral). These descriptions do not come from the result, thus please provide the logical support.

Authors’ comments:

Thank you for your comment. We acknowledge that, in the manuscript, the terms "factor" and "category," which were used in our previous study, were mixed with the term "factor," which was used because of the factor analysis in this study, making the manuscript very difficult to understand. We have revised the expressions in the manuscript to make it easier to distinguish the content of our previous study and this study. Furthermore, we added the details of findings obtained from the previous study to the Methods section to facilitate the reader's understanding.

P8, L148–

The qualitative study that we previously conducted [10] revealed that the motivation of stroke patients admitted to KRWs was influenced by two factors—personal and social-relationship factors. Four categories of personal factors (patients' goals, experiences of success and failure, physical condition and cognitive function, and resilience) and three categories of social-relationship factors (influence of rehabilitation professionals, relationship between patients, and patients' supporters) were included. Furthermore, the motivational status of stroke patients was shown to influence their behaviors, such as frequency of self-training and attitude toward activities in daily life. After referring to these previous findings [10], and the findings in another study on patients’ views regarding motivation for rehabilitation [19], four rehabilitation professionals and the authors of the study [three occupational therapists (TY, MK, and SK) and a medical doctor (YO)] discussed and created an item pool in Japanese for the MORE scale.

P23, L360–

All MORE scale items were developed by referring to two types of factors (personal and social-relationship factors) that influence patients’ motivation for rehabilitation and the content of motivated behavioral change [10].

Minor comment 1

There is inequity on the introduction of the various outcomes. Thus, there are too much expression about MOT-Q.

Authors’ comments:

Thank you for your suggestion. I have reviewed the amount of description of other motivational scales in the introduction and summarized the description of MOT-Q, in particular, as follows.

P5, L70–

Several scales for motivation for rehabilitation have been proposed for patients undergoing rehabilitation. However, each of these scales have some shortcomings. In self-determination theory [4,5], widely known as the motivation theory, motivation is broadly classified into intrinsic motivation and extrinsic motivation. Extrinsic motivation involves performing a particular activity because it leads to a separable consequence; that is, the goal is separate from the activity itself. In contrast, intrinsic motivation involves performing a particular activity because it is interesting and enjoyable [4,5]. According to this classification, rewards, including functional recovery and praise from medical staff and family members, can be categorized as extrinsic motivation, and patients' enjoyment of the rehabilitation itself can be classified as an intrinsic motivation. The motivation for traumatic brain injury rehabilitation questionnaire (MOT-Q) [6-9] consists of four extrinsic factors for patients’ motivation. However, it has been reported that stroke patients' motivation for rehabilitation is influenced by broader factors other than those assessed using the MOT-Q [10]. Thus, MOT-Q may overlook some aspects of the factors that may affect motivation. 

Minor comments 2 and 3

L97: based on “various categories”. Please specify the various categories.

L98 from medical staff but also narratives from stroke patients themselves [14]. It is difficult to understand the contents of this sentences and the reference 14.

Authors’ comments:

Thank you for your suggestion. The term "various categories" refers to categories that influence the motivation of stroke patients toward rehabilitation, and the contents of the behavioral changes, as identified in our previous study. The manuscript had inadequately explained the contents of the influencing categories identified in that study. Therefore, we have revised the manuscript as follows. Furthermore, as mentioned in answer to Major comment 5, we described the details about the findings obtained from the previous study in the Methods section.

P7, L112–

In this study, we developed the Motivation in stroke patients for Rehabilitation scale (MORE scale) by referring to two types of factors (personal and social-relationship factors). They influence the patients’ motivation for rehabilitation and the content of motivated behavioral change, which were revealed in our previous study [10].

Minor comment 4

L123: “from among” is strange to be read. Please fix it.

Authors’ comments:

Thank you for your suggestion. We deleted “among” from the text.

Minor comment 5

L128: “the item response theory” seems to be better to be revised into “item response theory analysis” and it applies on the remaining sentences on this manuscript.

Authors’ comments:

Thank you for your suggestion. We have revised the description from "the item response theory" to "item response theory analysis" throughout the manuscript.

Minor comment 6

The authors mentioned “Intensive rehabilitation”, however it is not defined in the manuscript, thus it needs operational definition

Authors’ comments:

Thank you for your comment. The participants of this study were those who were hospitalized in the Kaifukuki Rehabilitation Wards (KRWs). In KRWs, as described in the manuscript, the patients undergo rehabilitation with therapists for around 2–3 hours every day. Intensive rehabilitation refers to around 2-3 hours of training conducted every day during the patients’ hospitalization in KRWs. We have revised the text of the study setting so that the reader can understand the content, as follows.

P8, L142–

In KRWs, patients undergo one-on-one intensive rehabilitation with therapists for around 2–3 hours every day.

Minor comment 7

L158, 159: The authors describe association between motivation and apathy, and apathy and depression. However, the goal of the present study was to find association of motivation and apathy, and motivation and depression. Thus those part needs to be revised.

Authors’ comments:

Thank you for your comment. I have revised the manuscript as stated in the response to Major comment 3, to convey the main idea more clearly.

Minor comment 8

L161: needs grammatical correction.

Authors’ comments:

The grammar was checked by a native speaker, and corrections have been made as follows.

P11, L222–

In addition, the VAS, which rates the subjective feelings of motivation, was used for evaluating the criterion validity since there lacked valid rating scales that could assess the motivation of stroke patients.

Minor comment 9

L167: It might be better to describe the reason of using SDS other than other outcome measures on the depression. Moreover, the reference on the validation and reliability of SDS is needed.

Authors’ comments:

Thank you for your comment. To examine convergent, discriminant, and criterion validity, this study adopted a self-rating assessment of evaluation method. Furthermore, SDS was adopted because its validity and reliability have been confirmed in stroke patients, it is widely used as a depression assessment scale, and it is easy and quick to evaluate considering the fatigue of the participants. We have revised the manuscript as follows. Furthermore, we have added the description of SDS validity and reliability (please see the response to Major comment 2).

P12, L243–

The AS and SDS were adopted because these were self-rating scales similar to the MORE scale. Furthermore, these were easy and quick assessments in consideration of the participants’ fatigue.

Minor comment 10

L183-186: It needs clarification and specification of the process grouping the items, why the whole items were grouped into personal factor, social relationship factor.

Authors’ comments:

Thank you for your comment. The items of the MORE scale were developed based on the results of our previous research on the factors that influence the motivation and behavioral changes caused by motivation in stroke patients. In addition to the descriptions added in method section (please see the response to Major comment 5), the manuscript has been revised as follows to avoid misleading readers.

P12, L255–

The MORE scale contains 17 items, which were based on the following categories in our previous research [10]: four items (1,2,3,4) regarding the patients’ goals, three items (11,12,13) regarding success and failure experiences, one item (14) regarding physical condition and cognitive function, two items (16,17) regarding resilience, four items (5,6,7,8) regarding the influence of rehabilitation professionals, one item (9) regarding relationships between patients, one item (10) regarding patients’ supporters, and one item (15) regarding patients’ behavior changes. In our previous study, patients’ goals, success and failure experiences, physical condition and cognitive function, and resilience were based on the personal factors that influenced patients’ motivation [10]. The influence of rehabilitation professionals, relationships between patients, and patients’ supporters were based on the social relationship factors that influenced patients’ motivation [10].

Minor comment 11

L187: Why the Likert scale is made up with 7-point scale? Please state the reason.

Authors’ comments:

Thank you for your comment. The optimal Likert scale has been known to comprise 7–10 points. Considering the fatigue of the participants, we adopted the least number of points in the scale out of this range, namely, seven points. 

P13, L266–

Each item of the MORE scale was evaluated using Likert scale. Considering the participants’ fatigue, we selected the seven-point scale, which is known as the minimum optimal number on a Likert scale [38]. It was rated as follows: 1, Strongly disagree; 2, Disagree; 3, Somewhat disagree; 4, Neither agree nor disagree; 5, Somewhat agree; 6, Agree; and 7, Strongly agree.

38. Preston CC, Colman AM. Optimal number of response categories in rating scales: reliability, validity, discriminating power, and respondent preferences. Acta Psychol. 2000;104(1):1-15. doi: https://doi.org/10.1016/S0001-6918(99)00050-5

Minor comment 12

Table 2:

Paretic side: 0 could be misunderstood , thus lesion side seems to be better to transfer the glimpse of the participants.

Authors’ comments:

Thank you for your comment. I have modified the description from paretic side to lesion side (please see Table 2).

Minor comment 13

Foot-tap test rather than foot-pat test

Authors’ comments:

Thank you for your suggestion.

It is undeniable that the behavior is foot-tap; however, since the original name of the SIAS test item is “foot-pat test,” we have used “foot-pat test” in this manuscript.

Minor comment 14

L 204: Please specify “it”

Authors’ comments:

Thank you for pointing this out.

We have changed the notation of "It" to "MORE scale" in this section.

P16, L288–

The Kaiser-Guttman rule suggested that the MORE scale has a single factor structure (Eigenvalue = 9.11, variance extracted = 53.6%).

Minor comment 15

P15: structural validity says the MORE scale is not indicative of acceptable model fit. Then, I guess more revision is needed in the process, however there was no trial to improve the validity.

Authors’ comments:

Thank you for your comment. As mentioned in the answer to Major comment 4, we have revised the description of CFA.

Minor comment 16

P20 L12-18: It is difficult to understand the authors’ purpose of the table 6 and the sentences. I hope the authors emphasize the meaning of the table.

Authors’ comments:

Thank you for your comment. The results presented in Table 6 and P20, L12 indicate that there were a certain number of participants who were motivated (or not) for rehabilitation even though they were rated positive (or negative) for apathy and depression using the AS and SDS. This result shows that the MORE scale can specifically evaluate motivation for rehabilitation, regardless of specific conditions such as depression or apathy. To make it easier for readers to understand the manuscript, we have only included the parts of Table 6 that needed emphasis and modified the manuscript as follows.

P20, L333–

Furthermore, we validated the properties of the MORE scale using the cutoff value of the AS and SDS, and the quartile of the MORE scale. Participants with MORE scale scores below the first quartile were considered to have relatively low motivation. Table 5 shows the results of the total score of the AS, SDS, and MORE scale. Among the participants, 11.9% of those who were evaluated as having a score more than the cutoff point for both AS and SDS (both apathy and depression positive) obtained scores that were more than the third quartile for the MORE scale. However, 6.5% of participants who were evaluated as having a score below the cutoff point for both AS and SDS (both apathy and depression negative) were evaluated as having scores below the first quartile for the MORE scale. Thus, the results show that there were a certain number of participants who scored below the cutoff point for both depression and apathy, but had low motivation for rehabilitation. Conversely, there were a certain number of participants who scored above the cutoff point for both depression and apathy, but had high motivation for rehabilitation.

Minor comment 17

Table 6 : abbreviation of IQR needs to be described with full terminology. The row name IQR needs to be revised to convey the meaning of 1st ~ 3rd IQR.

Authors’ comments:

Thank you for your comment. We have corrected the manuscript as you suggested.

Minor comment 18

Table 5: It seems to be strange to correlate AS total score, SDS total score, and VAS with each item of MORE scale. Please specify the reason of this process.

Authors’ comments:

Thank you for your comment. To make it easier for readers to understand, Table 5 was deleted and only the correlations between the total score on the MORE scale and the average scores of the AS, SDS, and VAS scores were retained in the manuscript.

Minor comment 19

Discussion

Unlike in our hypothesis, this study's factor analysis, a one-factor structure was indicated.; In the introduction and method section, hypothesis about the factor is not found. Please describe it.

Authors’ comments:

Thank you for your comment. We have added a description of the hypothesis stated in the discussion to the introduction.

P7, L112–

In this study, we developed the Motivation in stroke patients for Rehabilitation scale (MORE scale) by referring to two types of factors (personal and social-relationship factors). They influence patients’ motivation for rehabilitation and the content of motivated behavioral change, which were revealed in our previous study [10].

P10, L196–

We hypothesized that the MORE scale would have a three-factor structure consisting of two motivational influence factors (personal and social-relationship), and a behavioral change factor, similar to the results of our previous qualitative studies [10].

Minor comment 20

“The category “goal setting,” which the items 1,2,3, and 4 referred to, have been reported to be related to the improvement of daily living activities [38,39]. "Pain" in Item 14, which is included in the category "physical condition", has been reported to have a negative effect on FIM improvement [40]. Similarly, the category "success experience", which are items 11, 12, and 13, and the category "resilience" (resilience against obstacles), which are items 16 and 17, were related to patients’ functional improvement [41,42]. Furthermore, it has been reported that the category "influence from supporters," including professionals and family members, in items 5,6,7,8,9, and 10, can affect patients' motivation for rehabilitation [15,43-46]. “These paragraphs seems to be strange, because the authors grouped each items and named it, without support by results. Lack of scientific evidence does not support the meaning of this article

Authors’ comments:

Thank you for your comment. The description is based on the results of a qualitative study on stroke patients that we conducted prior to developing the MORE scale. A summary of the content of our previous study would make it easier for the reader to understand this manuscript since much of the content of this manuscript was determined with reference to our previous qualitative research results. 

Therefore, we have added the summary of the findings obtained from the previous study to Method section (please see the response to Major comment 5).

Minor comment 21

“This study also investigated the cutoff point of the MORE scale”. There seems no description about the cutoff point of the MORE scale in the result section. Please amend it.

Authors’ comments:

Thank you for pointing this out. The cutoff point is not included in the current version of the manuscript. In the manuscript, the quartiles were used. We have revised the manuscript as follows.

P25, L401–

 This study also examined whether the study participants were motivated with reference to the quartiles of the MORE scale.

Minor comment 22

“The results showed that the MORE scale can evaluate patients’ motivation for rehabilitation specifically, regardless of depression and apathy.” needs evidence from results. If there is relevant result from this study, please describe it and explain with explanation.

Authors’ comments:

Thank you for your comment. The MORE scale results show that 6.5% of the participants who tested negative for both depression and apathy had relatively low motivation for rehabilitation. Furthermore, 11.9% of the participants who tested positive for both depression and apathy had scores that were more than the third interquartile range for the MORE scale. These results provide evidence that the MORE scale could evaluate motivation for rehabilitation, regardless of pathological conditions such as depression and apathy, as described in the discussion. To make this content more understandable, the manuscript has been revised as described in response to Minor comment 16.

---

## [Decision Letter · Decision Letter 1]

10 Feb 2022

PONE-D-21-29804R1Development and validation of new evaluation scale for measuring stroke patients’ motivation for rehabilitation in rehabilitation wardsPLOS ONE

Dear Dr. Otaka,

Thank you for submitting your manuscript to PLOS ONE. After careful consideration, we feel that it has merit but does not fully meet PLOS ONE’s publication criteria as it currently stands. Therefore, we invite you to submit a revised version of the manuscript that addresses the points raised during the review process. You will note that there are only comments from one reviewer, which is because I am happy to accept the original reviewer 1's recommendation. The final comments are located at the end of this email, but can be summarised as clarifying the abstract in relation to the objective and methods (relating to reliability and validity). I appreciate how difficult it is to keep the abstract within the 300 word limit; my suggestion would be to remove the first two sentences of the objective section as these can be described as background rather than a statement of objectives. This would also help to address the comment about needing to tone down the 'no reliable tools'. Of course you don't have to follow my suggestion, but please do address the comments. 

We look forward to receiving your revised manuscript.

Kind regards,

Jason Scott

Academic Editor

PLOS ONE

Journal Requirements:

Reviewers' comments:

Reviewer's Responses to Questions

**Comments to the Author**

1. If the authors have adequately addressed your comments raised in a previous round of review and you feel that this manuscript is now acceptable for publication, you may indicate that here to bypass the “Comments to the Author” section, enter your conflict of interest statement in the “Confidential to Editor” section, and submit your "Accept" recommendation.

Reviewer #2: All comments have been addressed

2. Is the manuscript technically sound, and do the data support the conclusions?

Reviewer #2: Yes

3. Has the statistical analysis been performed appropriately and rigorously? 

Reviewer #2: Yes

4. Have the authors made all data underlying the findings in their manuscript fully available?

Reviewer #2: Yes

5. Is the manuscript presented in an intelligible fashion and written in standard English?

Reviewer #2: Yes

6. Review Comments to the Author

Reviewer #2: I really appreciate for the detailed and elaborative answer for my recommendation.

The main article is excellent enough to be accepted.

I have few recommendation for the quality improvement in abstract, as follows

"No valid and reliable tools~": I hope the authors tone down this sentence. ex)It lacks of study about-

Results:

-There is no sentences about reliability, thus please add contents about reliability.

-The readers might have difficulty to find results relevant to convergent, discriminant and criterion validity. Thus please add more explanation in results for those validity.

7. PLOS authors have the option to publish the peer review history of their article (what does this mean?). If published, this will include your full peer review and any attached files.

Reviewer #2: **Yes: **Joon-Ho Shin

---

## [Author Response · Author response to Decision Letter 1]

16 Feb 2022

Reviewer Comments:

Comments from Reviewer #2:

I have few recommendations for the quality improvement in abstract, as follows

"No valid and reliable tools~": I hope the authors tone down this sentence. ex) It lacks study about-

Results:

-There is no sentences about reliability, thus please add contents about reliability.

-The readers might have difficulty to find results relevant to convergent, discriminant and criterion validity. Thus please add more explanation in results for those validity.

Authors’ comments:

Thank you for your constructive suggestions. To tone down the wording of the objective, the first two sentences of the objective section in the Abstract have been removed, because they would be considered background, rather than statement of the objective. We have added a description of the reliability to the results section and modified the wording in the text to make it easier for readers to understand convergent, discriminant, and criterion validity. We have revised the abstract as follows.

P3, L27-

Objective: This study aimed to develop the Motivation in stroke patients for rehabilitation scale (MORE scale), following the Consensus-based standards for the selection of health measurement instruments (COSMIN).

P3, L29-

Method: Study participants included rehabilitation professionals working at the convalescent rehabilitation hospital and stroke patients admitted to the hospital.

P3, L38- 

Results: Using the Delphi method, 17 items were incorporated into the MORE scale. According to EFA and CFA, a one-factor model was suggested. All MORE scale items demonstrated satisfactory item response, with item slopes ranging from 0.811 to 2.142, and item difficulty parameters ranging from -3.203 to 0.522. Cronbach’s alpha was 0.948. Regarding test-retest reliability, a moderate correlation was found between scores at the beginning and one month after hospitalization (rho = 0.612. p < 0.001). The MORE scale showed significant correlation with AS (rho = -0.536, p < 0.001), SDS (rho = -0.347, p < 0.001), and VAS (rho = 0.536, p < 0.001), confirming the convergent, discriminant, and criterion validity, respectively.

---

## [Editor Report · Decision Letter 2]

28 Feb 2022

Development and validation of new evaluation scale for measuring stroke patients’ motivation for rehabilitation in rehabilitation wards

PONE-D-21-29804R2

Dear Dr. Otaka,

We’re pleased to inform you that your manuscript has been judged scientifically suitable for publication and will be formally accepted for publication once it meets all outstanding technical requirements.

Kind regards,

Jason Scott

Academic Editor

PLOS ONE
---

## [Editor Report · Acceptance letter]

7 Mar 2022

PONE-D-21-29804R2 

Development and validation of new evaluation scale for measuring stroke patients’ motivation for rehabilitation in rehabilitation wards 

Dear Dr. Otaka:

I'm pleased to inform you that your manuscript has been deemed suitable for publication in PLOS ONE. Congratulations! Your manuscript is now with our production department. 

Kind regards, 

on behalf of

Dr. Jason Scott 

Academic Editor

PLOS ONE